# Image Analysis in Digital Pathology Utilizing Machine Learning and Deep Neural Networks [note 1]

**DOI:** 10.3390/jpm12091444

**Published:** 2022-09-01

**Authors:** Paris Amerikanos, Ilias Maglogiannis

**Affiliations:** Department of Digital System, University of Piraeus, 18534 Piraeus, Greece

**Keywords:** machine learning, computer vision, digital pathology, object detection, instance segmentation, breast cancer

## Abstract

Detection of regions of interest (ROIs) in whole slide images (WSIs) in a clinical setting is a highly subjective and a labor-intensive task. In this work, recent developments in machine learning and computer vision algorithms are presented to assess their possible usage and performance to enhance and accelerate clinical pathology procedures, such as ROI detection in WSIs. In this context, a state-of-the-art deep learning framework (Detectron2) was trained on two cases linked to the TUPAC16 dataset for object detection and on the JPATHOL dataset for instance segmentation. The predictions were evaluated against competing models and further possible improvements are discussed.

## 1. Introduction

The large amounts of raw data gathered in the medical field have allowed experts to access such data on individual patients, but the sheer amounts are simply impossible for humans to process, use or understand at a large scale. As a response to that challenge, statistical (SL), machine (ML) and deep (DL) Learning techniques are used in combination with big data frameworks to detect patterns and trends in that raw data and yield insights that are useful to experts.

Tasks to which ML is applied are numerous and wide-ranging; identifying potential drug combinations, assisting the health practitioner’s diagnosis using the available symptom data, or classifying medical images to the correct disease classes are some of the challenges ML is tasked to overcome. While specialized software and algorithms have been around for some time across most of the medical fields, the differences are substantial. Until recently, feature extraction was a rigid process, and the decision criteria were usually based on external research that could not be representative of actual use cases, as environment and populations could differ. On the other hand, ML algorithms can be dynamically updated with newer data provided by the expert consulting the software, providing increased accuracy of the produced diagnosis.

In the last decade, the development of information systems for the automated diagnosis of medical images constitutes a field of ever-growing scientific research. Digital medical images are present in most diagnostic labs, providing easy manipulation through various information systems. The digital processing of medical images by multiple feature extraction techniques can lead to the accumulation of numerous features in a reliable and reproducible way. The analysis of biomedical images through extracted features is a process that can be carried out by ML algorithms and ultimately augments the decision-making processes of medical experts by providing automated diagnosis insights. The information gain of such systems is significant, as it enhances the timely and reliable identification of important patient cases. Systems such as these can be incorporated into local information systems at diagnostic centers but can also be a part of a telehealth information system.

The motivation for this paper is the need for further developments in ML (machine learning) and CV (computer vision) algorithms in the medical field, and the transfer of those into production to assist with the processing of the large amount of available data. With the great advances in computational power achieved during the last decade, DL algorithms could be used to harness this power in specific medical tasks, thereby achieving better-than-human performance and significant time and cost savings. In this report, we decided to explore the detection of ROIs in WSIs, as it is a highly subjective and labor-intensive task, making use of the openly available TUPAC16 and JPATHOL datasets on breast cancer. Training and evaluation are based on a state-of-the-art deep learning framework (Detectron2), which tackles a variety of important CV tasks (e.g., object detection, instance segmentation) and iterates quickly through different architectures and parameters. Finally, the predictions are evaluated against competing approaches developed around the same datasets, and further possible improvements are discussed for use in production.

This paper opens with a brief introduction on deep learning in digital pathology in Section 2. The methodology and the two datasets used for model training are presented in Section 3. Section 4 presents the findings, and Section 5 presents the results and possible improvements.

## 2. Related Work and Background Information

### 2.1. The Concept of Digital Pathology (DP)

Over the last decade, the nature of diagnostic healthcare has changed rapidly, owing to an explosion in the availability of patient data for disease diagnosis [1]. Traditional methods of analysis of cancer samples were limited to a few variables and the measurement of a few clinical markers; the pathologist was trained to synthesize this information into a diagnosis that would help the clinician determine the best course of therapy. With the advent of cost-effective whole slide digital scanners, tissue histopathology slides are now stored in digital image form. The availability and analysis of much larger sets of variables, combined with sophisticated imaging and analysis techniques, are replacing the traditional paradigm of a pathologist and a microscope with a digital pathologist and a screen panel, where he or she can view and analyze digitized tissue sections.

Some paradigm-shifting developments in the field of DP are examined below.

**Diagnosis**: Dramatic increases in computational power and improvement in image analysis algorithms have allowed the development of powerful computer-assisted analytical and ML approaches to biomedical data, assisting in more accurate diagnoses. Pathologists can now spend more time and resources analyzing patient edge cases where nonstandard features may be present. Consequently, several researchers have begun to develop computer-aided diagnosis methods by applying image processing and computer vision techniques to try and identify spatial extent and location of diseases on digitized tissue sections.

**Prognosis**: Another task of DP is to identify prognostic markers and predict disease outcome and survival. For instance, breast cancer grade is known to be highly correlated to patient outcome and long-term survival.

**Theragnosis**: Some diseases, e.g., cancer, are complex and not yet fully understood. The same treatment applied to two patients with apparently similar diseases may have significantly different outcomes. This may be patient-specific, but a part is also due to our limited understanding of the relationship between disease progression and clinical presentation. There is a consensus among clinicians and researchers that a more detailed approach, using computerized imaging techniques to better understand tumor morphology, combined with the classification of diseases into more meaningful molecular subtypes, will lead to better patient care and more effective therapeutics. The variables that can be used in such an analysis are the tumor molecular features, results from the imaging of the tumor cellular architecture and microenvironment, the tumor 3D tissue architecture and vascularization and its metabolic features. While DP offers very interesting, highly dense data, one of the exciting challenges in the future will be in multimodal data fusion for making therapy recommendations (theragnosis), especially with regard to personalized medicine.

**Outlook**: Predictive, preventive, and personalized medicine will transform medicine by decreasing morbidity in cancer. This transformation will be driven by the integration of multiscale heterogeneous data. The goal of many scientists is a future where disease diagnostics will involve the quantitative integration of multiple sources of diagnostic data, including genomic, imaging, proteomic and metabolic data that can distinguish between individuals or between subtle variations in the same disease to guide therapy. Quantitative cross-modal data integration will also allow disease prognostics, enabling physicians to predict susceptibility to a specific disease, as well as disease outcome and survival. Finally, the analysis will provide the ability to predict how an individual will react to various treatments; such a theragnostic profile would be a synthesis of various biomarkers and imaging tests from different levels in the biological hierarchy. A collection of these profiles, followed up over time, would provide insights into the disease process and be useful for improvements in developing future treatment options.

**Role of the pathologist**: The primary purpose of the ML tools mentioned previously is to complement the role of the pathologist. Most histopathology image analysis researchers are computer vision researchers and, as such, it is important to maintain a constant collaboration with clinical and research pathologists throughout the research process. There are unique challenges to the analysis of histopathology imagery, particularly in the performances required for eventual use of the technique in a clinical setting. It is the pathologist who can best provide feedback on the system’s performance, as well as suggest new avenues of research that would provide beneficial information to the pathologist community. Additionally, it is the pathologist who is best equipped to interpret the analysis results by considering the underlying biological mechanisms, which, in turn, may lead to new research ideas.

In this paper, the use of a specific deep learning platform is explored with the implementation of object detection and image segmentation tasks in the field of digital pathology. The datasets, tools and techniques used are focused on breast cancer—which is the leading type of cancer in women—but they can be extended to various other types of cancer that involve the same tissue histopathological data.

Breast cancer accounts for 25% of all cancer cases in women worldwide. In 2018, it resulted in 2 million new cases and 627,000 deaths [2]. It is more common in developed countries and is more than one-hundred times more common in women than in men. The five-year survival rates in the United Kingdom and the United States are between 80 and 90%. The diagnosis of breast cancer is confirmed by taking a biopsy of the concerning tissue, and, once the diagnosis is made, further tests are carried out to determine if the cancer has spread beyond the breast and which treatments are most likely to be effective.

The following section presents the prominent computer-based methodologies and tools that may assist a digital pathologist in analyzing and diagnosing cancer cases with greater accuracy and speed.

### 2.2. State-of-the-Art Computer-Based Methodologies and Tools for Digital Pathology

#### 2.2.1. Computer Vision

Computer vision (CV) is a research field that studies how computers can gain high-level understanding from digital images or videos [3,4], and how to automate tasks that the human visual system can do, such as extraction, analysis, and understanding of useful information from images. It involves the development of a theoretical and algorithmic basis to achieve automatic visual understanding [5]. The image data can take many forms, such as sequences, multi-views, or multi-dimensional data from medical machines.

CV feature-based methods are used in conjunction with ML/DL techniques and complex optimization frameworks [6]. The accuracy of DL algorithms on several benchmark data sets has allowed significant progress in tasks including, but not limited to, image classification, object localization, instance segmentation and keypoint detection. The principles of CV are similar to ML tasks. First, a suitable representation of the content of a digital image is pursued so that features can be extracted. Second, a model is trained on those features and produces a prediction on the new unseen content. Common techniques for feature extraction are eigenfaces and histograms of oriented gradients (HOG).

#### 2.2.2. Machine Learning

ML is a research branch of artificial intelligence that allows for the extraction of meaningful patterns from examples, just as human intelligence allows [7]. However, unlike humans, a computer can cover a variety of use cases, as it will perform a given repetitive task consistently and efficiently. In the last few decades, computers have demonstrated the ability to learn and even have become proficient in tasks that were thought to be too complex for machines, showing that ML algorithms can be critical components in decision support systems. Moreover, in some cases, computers have been found to discern patterns imperceivable to humans [8]. Naturally, this has led to heightened enthusiasm in the field of ML, especially along with the latest substantial increases in computational performance and available data.

ML algorithms generally fall into four broad categories, each of which has its own applications. Supervised learning relates inter alia to optical character recognition, speech recognition, image classification, language translation, sequence generation and object detection. Dimensionality reduction and clustering are categories of unsupervised learning, and self-supervised learning uses heuristic algorithms to generate synthetic labels for the data. Reinforcement learning has applications in the field of self-driving cars, robotics, resource management, education, etc. Even though some of those applications were unattainable up until recently, state-of-the-art advances have made them feasible.

Another significant application of ML could be found in medical imaging. Using ML algorithms to perform detection and diagnosis can help medical experts interpret medical imaging findings and reduce analysis times [9,10]. Examples of medical tasks where such algorithms have been used are pulmonary embolism segmentation with CT-angiography, polyp detection with virtual colonoscopy, breast cancer detection with mammography, brain tumor segmentation with magnetic resonance (MR) imaging, etc.

#### 2.2.3. Feature Extraction and Selection

The primary process in ML is the retrieval of features that contain the information on which insights will be based, usually performed by assigning quantitative values to visual textures [11,12]. Even though visual feature learning is easy for humans, computing and representing features is a complex task. Visual features must be robust enough to overcome morphological variations including, but not limited to, rotations, noise, and intensity differences.

It is possible to retrieve a great number of features from a given image, but too many features can stunt the learning process. Feature selection avoids overfitting by selecting only a subset of the features, often by looking for correlations among them; a large number of correlated features may mean that some features can be omitted with minimal information loss. The needed features that will allow a model to differentiate among the various classes require a minimum number of samples, which depends mainly on the distinctness of each class.

There is a variety of ML algorithms available for obtaining the optimum features, based on differing data assumptions, and depending on feature adjustment methods. Some of the most commonly used ML algorithms are k-nearest neighbors, support vector machines, decision trees, naïve Bayes classifiers and neural networks.

By the current definition, deep learning (DL) is a branch of ML and concerns neural networks with multiple layers between the input and output layers. Due to limitations in computing power and difficulties in the backpropagation process, most primitive neural networks had less than three layers [13]. These challenges have been mostly overcome through leveraging the power of parallel computing GPUs and different neural network architectures, such as stacked auto encoders, recurring neural networks (RNNs) and convolutional neural networks (CNNs), etc. Concerning the DNN’s architecture, there are no set rules to define the correct number, type, size or ordering of layers for a given problem—it is still a trial-and-error process.

#### 2.2.4. Deep Learning in Computer Vision

Rapid progressions in the DL field and improvements in device capabilities have improved the performance and cost-effectiveness of vision-based applications. Compared to traditional CV techniques, DL methods achieve greater accuracy in tasks, such as image classification, semantic segmentation, and object detection [14]. Since neural networks used in DL are trained rather than programmed, applications using this approach often require less expert analysis and fine-tuning and can exploit the tremendous amount of data available today. DL also provides superior flexibility, because CNN models and frameworks can be re-trained using a custom dataset for any use case.

The traditional approach in image analysis is the use of well-established CV techniques, such as feature descriptors (SIFT, SURF, BRIEF, etc.) that extract features, small descriptive image patches, through algorithms such as edge/detection or threshold segmentation. The difficulty with this traditional approach is that it is necessary to manually select the important features in each given image. DL introduced the concept of learning, where the machine is presented with a dataset of images that have been annotated with a ground truth and it automatically discovers the underlying patterns, and automatically works out the most descriptive features, with respect to each specific class for each object. It has been well-established that DNNs perform far better than traditional algorithms, albeit with trade-offs regarding computing requirements and training time [15].

The development of convolutional neural networks (CNN) has had a tremendous influence and is responsible for significant improvements in the field of CV and object recognition. CNNs are mainly used in CV tasks, because they make the explicit assumption that the inputs are images, encoding certain properties into the architecture. Unlike regular NNs, the layers of a CNN have neurons arranged in three dimensions, width, height, depth. The neurons in a layer will only be connected to a small region of the layer preceding it, instead of all the neurons in a fully connected manner. Moreover, in the final output layer, the full image will be reduced to a single vector of class scores arranged along the depth dimension. The most common form of a CNN architecture is a linear list of layers that stacks a few convolutional layers, follows them with pooling layers, and repeats this pattern until the image has been merged spatially into a small size. At some point, it is common to transition to a fully connected layer as an output layer.

CNN architectures, such as Google’s Inception and Microsoft’s ResNet, feature more intricate and different connectivity structures. Further commonly used CNN architectures are LeNet, AlexNet, ZF Net, GoogLeNet and VGGNet [16,17]. In most practical real-world applications, instead of developing novel architectures for a problem, a pre-trained model of whatever CNN architecture works best on datasets, such as ImageNet, can be used and finetuned on the required data.

#### 2.2.5. Existing Approaches vs. Deep Learning

Digital pathology is becoming increasingly common due to the growing availability of whole slide digital scanners. The digitized slides allow the use of image analysis techniques in detection, segmentation, and classification. Algorithmic approaches have proven to be beneficial, as they have the capacity to significantly reduce the laborious and tedious nature of providing accurate quantifications, as well as reduce inter-reader variability among pathologists [18]. Several image analysis tasks in DP involve some sort of quantification or tissue grading and invariably require identification of histologic primitives (e.g., nuclei, mitosis, tubules, epithelium, etc.).

As a result, there is a strong need to develop efficient and robust algorithms for DP image analysis. While there have been a few papers in computational image analysis for the purposes of object detection and quantification [19,20], there appear to be two main drawbacks to the existing approaches. First, the development of task-specific approaches tends to require long research and development cycles; an algorithmic scheme needs to be developed that can account for as many of the variances as possible, while not being too general to avoid the result of false positives or too narrow to avoid the result of false negative errors. This process can become quite unwieldy as a priori, it is often unfeasible to view all the outlier cases and, thus, an extensive iterative trial-and-error approach needs to be undertaken. The second drawback is that the required, but also limited, implicit knowledge of how to find or adjust optimal parameters is not intuitively understood by external parties other than the developers. Together, these create a strong hindrance for researchers to leverage or extend the available technology to investigate their clinical hypothesis.

DL is an example of the ML paradigm of feature learning, wherein it iteratively improves upon learned representations of the underlying data with the goal of maximizing class separability. There are no preexisting assumptions that guide the creation of the learned representation; this approach involves deriving a suitable feature space solely from the data itself. This is a critical attribute of the DL family of methods, as learning from training exemplars allows for the generalization of the learned model to other independent test sets. Once the DL network has been trained with an adequately powered training set, it is usually able to be effectively generalized to unseen situations, countering the need of manually engineering features.

DL is suited to analyze big data repositories, where employing a feature engineering approach would require several algorithmic iterations and substantial effort to capture a similar range of diversity. Many manually engineered or feature-based approaches are not implicitly poised to manipulate and distil large datasets into classifiers in an efficient way. DL approaches, on the other hand, have the potential to become the unifying approach for the many tasks in DP, having previously been shown to produce state-of-the-art results across varied domains [7,18]. As such, the focus of this manuscript is to discuss the usage of a single framework that can be tweaked to apply to a diverse set of unique use cases.

## 3. Material and Methods

### 3.1. Methodology and Implementation

The use cases examined in this project, such as object detection of mitotic figures in TUPAC16 [2] and instance segmentation of histologic primitives (Nuclei, Epithelium, Tubules) in JPATHOL [18], demonstrate how DL can be applied to a variety of the most common image analysis tasks in DP. For this purpose, the open-source DL framework Detectron2 [21] is leveraged, using the Mask R-CNN architecture pretrained on the COCO dataset [22].

The ultimate objective is to investigate whether a single training and model-building paradigm can be applied to each task, solely by determining and modifying its hyperparameters, but yet being able to generate results that are comparable to competing DL solutions or better than handcrafted approaches. This convergence to a unified approach not only allows for a low maintenance overhead, but also implies that image analysis researchers or DP users face a minimal learning curve, as the overall learning paradigm and hyperparameters remain constant across all tasks.

#### Deep Learning Framework Specifications

While searching for a single DL framework that could run both required tasks, Facebook AI Research released Detectron2, a next-generation platform for object detection and instance segmentation. It is a ground-up rewrite of Detectron, it is powered by PyTorch, and trains faster than its originator Mask R-CNN-benchmark [23].

Instance segmentation combines localization and classification of individual objects in an image (object detection) and classification of each pixel in an image (semantic segmentation). Mask R-CNN extends Faster R-CNN [24] by adding a parallel branch for predicting segmentation masks on each RoI, in parallel with the existing branch for classification and bounding box regression. This backbone branch uses the ResNet family of CNNs. It adds only a small computational overhead, but surpasses all previous state-of-the-art single-model results on the COCO object detection and instance segmentation tasks.

The Detectron2 framework was set up on a system running Linux 18.04 and Python 3.6. The system running the setup is comprised of a 6-core 3.00 GHz CPU, 16 MB 3000 MHz DDR4 RAM and an Nvidia RTX2070 8 GB GPU.

### 3.2. Use Cases Description

#### 3.2.1. Object Detection on TUPAC16

Tumor proliferation is an important biomarker that is indicative of the prognosis of breast cancer patients. Patients with high tumor proliferation have worse outcomes compared to patients with low tumor proliferation [25]. The assessment of tumor proliferation influences the clinical management of the patient; patients with aggressive tumors are treated with more intrusive therapies, and patients with indolent tumors are given more conservative treatments that are preferred because of fewer side effects [26].

Tumor proliferation in a clinical setting is traditionally assessed by pathologists. The most common method is to count mitotic figures (dividing cell nuclei) on hematoxylin and eosin (H&E) histological slides under a microscope. The pathologists will assign a mitotic score of 1-2-3, with increasing tumor proliferation. Although mitosis counting is routinely performed in most pathology practices, this highly subjective and labor-intensive task suffers from reproducibility problems [2]. One solution is to develop automated computational pathology systems to detect and count mitotic figures on histopathological images efficiently, accurately and reliably.

The first challenge on the topic of mitosis detection was MITOS 2012, hosted at the International Conference of Pattern Recognition (ICPR) [27]. In 2013, Veta et al. organized AMIDA13 in conjunction with the International Conference on Medical Image Computing and Computer Assisted Intervention (MICCAI) [28]. Mitosis detection was also one of the tasks of the MITOS-ATYPIA-14 challenge, organized as part of ICPR 2014, with the other task being the scoring of nuclear atypia [27]. A limitation of the previous challenges was that they focused solely on mitosis detection in predetermined tumor ROIs. However, in a real-world scenario, automatic mitosis detection is performed in WSIs, and an automatic method should ideally be able to produce a breast tumor proliferation score, with a WSI as the input. To address the above problem, [2] the Tumor Proliferation Assessment Challenge 2016 was organized for the prediction of tumor proliferation scores from WSIs.

The challenge included the following three main tasks to predict tumor proliferation: mitotic score prediction, gene expression-based PAM50 proliferation score prediction and ROI and mitosis detection. The latter concerns the design of a WSI tumor proliferation scoring system by identifying ROIs and counting mitoses, similar to how a pathologist would assess a slide for tumor proliferation.

Mitosis detection is the task being examined, and its dataset consists of WSIs from 73 breast cancer cases from 3 pathology centers, with annotated mitotic figures by the consensus of 3 observers. Of the 73 cases, 23 were previously released as part of the AMIDA13 challenge [28]. These cases were collected from the Department of Pathology at the University Medical Center in Utrecht, Netherlands. Each case was represented with varying numbers of HPFS, extracted from WSIs acquired with the Aperio ScanScope XT scanner at 40× magnification, with a spatial resolution of 0.25 μm/pixel. The remaining 50 cases previously used to assess the inter-observer agreement for mitosis counting were from 2 other pathology centers in the Netherlands (Symbiant Pathology Expert Center, Alkmaar and Symbiant Pathology Expert Center, Zaandam) [2]. Each case was represented by one WSI region, with an area of 2 mm^2^. These WSIs were obtained using the Leica SCN400 scanner (40× magnification and spatial resolution of 0.25 μm/pixel). In total, the mitosis detection auxiliary dataset contained 1552 annotated mitotic figures. Of the 656 provided images, only 587 of those that contained the annotated mitotic figures mentioned previously were used for training and validation; these images contained between 1 and 67 mitotic figures each.

The top scoring method for the mitosis detection task had an F-score of 0.652, which is a slight improvement over the top scoring method of AMIDA13 challenge with an F-score of 0.612.

#### 3.2.2. Instance Segmentation on JPATHOL

The JPATHOL paper [18] presents seven use cases that represent the ensemble of the components necessary for most of the current challenges at the DP image analysis stage, each with its own corresponding datasets. This report’s focus shall be on the segmentation tasks, where the delineation of accurate boundaries for histologic primitives (nuclei, epithelium, tubules) is required to extract precise morphological features. The training and tuning of a DL model shall be carried using all three datasets both separately and simultaneously, as observed in Table 1.

The ground-truth annotations are usually performed by an expert who delineates the object boundaries or annotates the pixels corresponding to a ROI. The level of annotation precision is critical in the optimization of supervised classification systems, but generating these annotations is often an arduous task due to the large amount of time and effort required. Pathologists are typically unavailable to perform the amounts of laborious manual annotations at the high resolutions needed for training and evaluating supervised ML algorithms. As a result, annotations are rarely pixel-level precise, and they are usually carried out at a lower magnification and tend to contain numerous false positives and negatives.

The DL network used for each of the individual tasks outlined in JPATHOL was a stock AlexNet architecture identical to the one Caffe provided. Its configuration is presented in Table 2, and its hyperparameters shown in Table 3 were held constant for all tasks to illustrate how parameter tweaking and tuning were not important in achieving good quality results. Experiments using dropout showed no improvement in the results, and lack of overfitting evidence dissuaded an approach using dropout.

Nuceli segmentation is an important problem, because nuclei configuration is correlated with outcome, and nuclear morphology is a key component in cancer grading schemes. Manually annotating all the nuclei in a single hematoxylin and eosin (H&E)-stained estrogen receptor positive breast cancer image is laborious and cannot be generalized to represent all the other variances present in other patients and their stain/protocol variances. As a result, time is better invested annotating sub-sections of each image, although this creates a challenging situation for generating training patches. Typically, one would use the annotations as a binary mask created for the positive class, and the negation of that mask as the negative class, randomly sampling from both to create a training set. In this case, while one can randomly sample from the positive mask successfully, the random sampling from the complement image may or may not return unmarked nuclei belonging to the positive class. To compensate, the standard approach is extended with intelligently sampled challenging patches for the negative class training set. Using a basic color deconvolution thresholding approach to select random negative patches, further segmented nuclei are obtained, even though the network is unable to identify nuclear boundaries accurately. To enhance these boundaries, an edge mask is produced by morphological dilation, and negative training patches are selected, which are inherently difficult to learn due to their similarity with the positive class. This patch selection technique results in clearly separated nuclei with more accurate boundaries.

Epithelium identification is important, since regions of cancer are typically manifested there. Work by [29] suggests that histologic patterns within the stroma might be critical in predicting overall survival and outcome in breast cancer patients. Thus, from the perspective of developing algorithms for predicting prognosis of disease, epithelium/stroma separation becomes critical. This task is unique in that it is less definitive than the more obvious tasks of mitosis detection and nuclei segmentation, where the expected results are quite clear. Epithelium segmentation, especially the subcomponent of identifying clinically relevant epithelium, is typically performed more abstractly by experts at lower magnifications.

Given that the AlexNet approach [18] constrains input data to a 32 × 32 window, the task is scaled to fit into this context. The principle is that a human expert should be able to make an educated decision based solely on the context present in the patch supplied to the DL network, which implies that a priori, an appropriate magnification must be selected from which to extract the patches and perform the testing. Networks with larger patch sizes could use higher magnifications at the cost of longer training times. Similar to nuclei segmentation, the objective is to reduce the presence of uninteresting training examples in the dataset, which can be carried out by applying a threshold to the grayscale image, thus removing fat and background pixels from the patch selection pool. In addition, to enhance the classifier’s ability to provide smooth boundaries, samples are taken from the outside edges of the positive regions.

Tubule segmentation can automate the area estimation with decreasing inter-/intra-reader variances and greater specificity, which can lead to better prognosis indication stratifications. Tubules are complex structures that consist of numerous components (e.g., nuclei, epithelium, lumen), which also determine their boundaries. In benign cases, tubules appear in a well-organized fashion with similar size and morphological properties, making their segmentation easier, while in cancerous cases, the organization structure breaks down and accurately identifying the boundary becomes challenging. In addition, tubules as an entity are much larger compared to the individual components. Thus, they require a greater viewing area to provide sufficient context.

In this use case, Janowczyk in 2016 [18] introduces the concept of using low-cost preprocessing to help identify challenging patches; per image, a random selection of pixels belonging to both classes to act as training samples is made, and a limited set of texture features (i.e., contrast, correlation, energy and homogeneity) is computed. Next, a naive Bayesian classifier determines the posterior probabilities of class membership for all the pixels in the image. In this manner, pixels that would potentially produce false positives and negatives are identified, and the training set is improved by removing trivial samples, without requiring any additional domain knowledge. Lastly, knowing that benign cases are easier to segment than malignant cases, patches are disproportionally selected from malignant cases to further help with generalizability.

Benign sections of tissue have stronger features and are more easily generalizable. On the other hand, malignant tubules are far more abstract and tend to have the hallmarks of a tubule, such as a clear epithelial ring around a lumen, which is less obvious, making them harder to generalize. This is potentially one of the downfalls of ML techniques that make inferences from training data; when insufficient examples are provided to cover all cases expected in testing, the approaches begin to fail. In this case, these challenges, in particular, could be addressed by providing a larger database of malignant images.

#### 3.2.3. Data Processing and Consumption

For standard tasks, such as object detection or instance segmentation, the standard representation for a dataset to be consumed by Detectron2 has a specification similar to COCO’s JSON annotations, which is a JSON file with a list of dictionaries (one per image), one for each subset of training, validation and testing data. Each image dictionary contains fields corresponding to image file paths, dimensions, unique IDs and a list of annotations for every instance featured in the image. Each annotation contains the bounding box coordinates, the label, and the segmentation mask of the instance, either as a list of polygons or as a per-pixel bitmap segmentation mask in COCO’s RLE format.

For TUPAC16, since the annotations are coordinate tuples instead of bounding boxes or polygons required for the object detection task, bounding boxes were created around each mitotic figure, with side dimensions set to 80 px. This initial value was set approximately by viewing several images with their annotations overlaid and ensuring that the bounding boxes contained a significant portion of the mitotic figures. At a later stage, the bounding box dimension was tuned as a model hyperparameter for maximum accuracy.

Another preprocessing task was the splitting of large images into smaller parts. As the input window for the model is 224 × 224 px, the initial image dimensions (2000 × 2000 px) would cause every image fed into the model to be resized, leading to significant loss of information and detail due to antialiasing. All images with dimensions greater than the CNN window were contiguously split into 224 × 224 px subimages and saved separately in the JPEG format, along with their corresponding annotations. After training/prediction, the subimages may be rejoined to recreate the original image along with their annotations. Any resulting subimages smaller than the CNN window were padded with black pixels to conform with the dataset. Subimages with no annotations were removed from the dataset, as they offered no significant knowledge, and the rest were split into training, validation, and testing subsets with a 60:20:20 ratio.

For JPATHOL, the first preprocessing step is to ensure that all datasets have the same magnification, so the epithelium images are zoomed in 2×, resulting in their new dimensions at 2000 × 2000 px. In addition, since the images in the tubule dataset do not have the same height and width, they are padded with black pixels in order to make them rectangular without affecting their aspect ratio or magnification, changing from 775 × 522 px to 775 × 775 px. The bitmap annotations undergo the same process in order to retain correct the ground-truth format.

The second step, similar to TUPAC16, involves splitting the images into smaller subimages that are analogous to the CNN window dimensions to avoid information loss. In this case, 261 px is selected for resizing as the least common denominator of the dimensions of the three data sets to reduce cases of subimages, consisting of mostly black padding. The third step is the preparation of segmentation masks for each instance and image. The ground truth for all three datasets is provided as 1- or 3-channel bitmap files, which are transformed into 1-channel binary arrays and are encoded into the COCO RLE format. Lastly, subimages with no annotations are discarded and the rest are randomly split into training, validation, and testing subsets with a 60:20:20 ratio.

#### 3.2.4. Parameterization and Training

After the images have been preprocessed and registered, they can be consumed by Detectron2 to train and evaluate an instance segmentation model. This model is based on Mask R-CNN with a ResNet R50-FPN backbone, pretrained on the COCO dataset (trained on Train2017 and evaluated on Val2017). In order to train the model on these datasets with the maximum possible accuracy, the model’s hyperparameters need to be tuned. The hyperparameters tuned in this research are presented in Table 4, along with the range of values on which they were tested, their initial/default values and their final values.

Finding the optimum values is a process that can be performed either manually or automatically. With the automatic approach, a script is prepared that iteratively trains models using all the consecutively or randomly chosen points in the hyperparameter grid, whose limits are suggested values found in the documentation or which are arbitrarily set within a reasonably expected scope. The downside of this approach is the time cost, as many models with hyperparameters far from the optimum values are needlessly trained. With the manual approach, models are iteratively trained, starting with random or suggested initial hyperparameter values and following a fashion of gradient descent towards the optimum values by changing one or two values per iteration. The downside of this approach is the possibility of converging to a local minimum in the hyperparameter space. Another approach is using Bayesian optimization approach, but this was not tested in this case, where the manual search approach was selected instead.

During the hyperparameter tuning phase, the resulting models are evaluated with the validation subset exclusively. For each hyperparameter value combination, the same model is trained and evaluated multiple times so that more precise mean and standard deviation values of each model’s accuracy can be calculated, mitigating the variance caused by the CNN’s stochastic nature. The evaluation stage returns twelve performance metrics (AP/AP50/APs/AR/etc.), as defined by COCO; the final decision process is based on the average precision for the whole area (AP), traditionally called mean average precision (mAP).

For TUPAC16, a major hurdle in histopathology image analysis is the variability in tissue appearance. The staining color and intensity can be significantly different between WSIs due to variation in tissue preparation, staining and digitization processes. To address this, for most similar tasks, staining normalization is performed as a preprocessing step. The most common method is the one proposed by [30], where an unsupervised method heuristically estimates the absorbance coefficients for the H&E stains for every image and the staining concentrations for every pixel. Afterwards, normalization is performed by recomposing the RGB images from the staining concentration maps using common absorbance coefficients.

This method was approximated by normalizing the RGB histograms of the images jointly and separately, but the color balance and visual features were greatly distorted. The resulting significantly lower accuracy compared to the initial unprocessed images led to this method being rejected after some trial runs on the TUPAC16 dataset.

For JPATHOL, the Detectron2 hyperparameters for this task were taken directly from the TUPAC16 task, as it offered a good baseline, and both datasets are similar in structure. Normalization was not performed, as it was not found to offer any improvements in accuracy. Dropout during training was also not performed as it showed no performance improvements and requires a smaller optimal dataset [31].

The most important element of this dataset is the experimentation required concerning patch generation. The generation of annotations for a dataset such as this is a cumbersome process, due to the large amount of time and labor needed. For example, the nuclei annotation dataset used in this task required over 40 hours to annotate its 12,000 nuclei, and yet it represents only a small fraction of the total number of nuclei present in all images. Unfortunately, this creates a challenging situation for generating training patches. Typically, the annotations would be used as a binary mask created for the positive class, and the negation of that mask as the negative class, then random sampling from both would be used to create a training set. In this case, however, while one can successfully randomly sample from the positive mask, the random sampling from the complement image may or may not return unmarked nuclei belonging to the positive class.

Consequently, extended image patches need to be generated to represent the available but unannotated ground truth more fully in the training and validation sets. This stage requires modest domain knowledge to ensure good representation of diversity in the training set. Selecting appropriate image patches for the specific task could have a dramatic effect on the outcome. Especially in the domain of histopathology, substantial variance can be present within a single target class, such as nuclei. This is especially pronounced in breast cancer nuclei, where nuclear areas can vary upwards of 200% between nuclei. Ensuring that a sufficiently rich set of exemplars is extracted from the images is perhaps one of the most key aspects of leveraging and utilizing a DL approach effectively.

For each of the three classes of images in JPATHOL, a detailed description of approaches is suggested that allows for the tailoring of training sets towards improving the specific detection tasks.

For nuclei segmentation, a standard approach involves selecting patches from the positive class and using a threshold on the color-deconvolved image to determine examples of the negative class. This rationale is based on the fact that non-nuclei regions tend to weakly absorb hemotoxin. The resulting network has very poor performance in correctly delineating nuclei, since these edges are underrepresented in the training set. This is compensated by extending it with intelligently sampled challenging patches for the negative class training set. Through the identification of positive pixels and the basic color deconvolution thresholding approach to select random negative patches, the segmented nuclei are obtained. However, the network may be unable to identify nuclear boundaries accurately, so an edge mask is produced by morphological dilation, where negative training patches are selected. A small proportion of the stromal patches is still included to ensure that these exemplars are well represented in the learning set. This patch selection technique results in clearly separated nuclei with more accurate boundaries.

For epithelium segmentation, similar to the nuclei segmentation task, the presence of uninteresting training examples in the dataset must be reduced, so that learning time can be dedicated to more complex edge cases. Epithelium segmentation can have areas of fat, or the stage of the microscope can be removed by applying a threshold to the grayscale image, thus removing those pixels from the patch selection pool. In addition, to enhance the classifier’s ability to provide smooth boundaries, samples are taken from the outside edges of the positive regions.

For tubule segmentation, a number of pixels per image belonging to both classes is randomly selected to act as training samples and compute a limited set of texture features. Next, a naive Bayesian classifier determines the posterior probabilities of class membership for all the pixels in the image, and pixels are identified that would potentially produce false positives and negatives and would benefit from additional representation in the training set. These pixels are selected based on their magnitude of confidence, such that false positives with greater posterior probabilities are more likely to be selected. This approach further helps to bootstrap the training set by removing trivial samples without requiring any additional domain knowledge. Lastly, knowing that benign cases are easier to segment than malignant cases, patches are disproportionally selected from malignant cases to further help with generalizability.

For experimental purposes, the problem was not treated as separate single-class instance segmentation tasks, but as a combined multi-class instance segmentation task, where the detector would need to both classify the input image to the correct class and detect and segment their instances. Unfortunately, due to lack of field expertise, the above processing suggestions could not be implemented on the subtasks. As this task is more of a proof of concept, actual performance is not considered to be the highest priority at this stage. Implementation of those suggestions, with or without the use of additional knowledge, is expected to increase the performance of the model dramatically to a level where it could raise interest for further research.

## 4. Experimental Results

The datasets and the performance results for the models trained for the two tasks are presented below. All the results presented are average precision metrics run on the validation subset (20% of initial dataset). The first experiments were run several times, but as the AP improved, multiple runs were made (up to 12 runs) per hyperparameter combination to achieve a more statistically precise value. For each of the final task models, a separate testing subset (20% of initial dataset) was used to measure their performance on previously unseen data.

### 4.1. TUPAC16

The TUPAC16 dataset contains 73 WSI images in PNG format, annotated by a list of text files containing the coordinates of the central points of each mitosis figure present in each of the images. In total, the dataset contains 1552 mitotic figures.

In Table 5, a few key hyperparameter value combinations are displayed, along with the AP their model achieved on the validation set. The average training time spent for each is also mentioned. In all, more than 40 hyperparameter combinations were used to train an equal number of models, and more than 170 runs were performed to validate these models.

The optimum combination of hyperparameters that resulted in the model with the highest average precision (65.02%) is shown in the last row of Table 5. For the testing subset, the average precision is 65.14%, which shows that overfitting was avoided. The average F-score for this model is $F_{1}=$ 0.628.

With regard to image predictions, a few random images from the test set are presented in Figure 1, with the ground-truth bounding boxes (left), juxtaposed with the bounding boxes predicted by the model (right).

### 4.2. JPATHOL

The JPATHOL dataset is comprised of three separate medical datasets featured in the work of Janowczyk [18], prepared for experimentation with nuclei, epithelium, and tubule segmentation tasks. Each dataset is comprised of images of different count, format, dimensions and magnification.

Table 6 presents the performance of the models trained on each of the nuclei/epithelium/tubule datasets separately, as well as the performance of the model trained on all of them simultaneously. All models were trained on the optimum hyperparameters found from the TUPAC16 task (Table 4), apart from *max iterations,* which had to be tuned independently for each to avoid disappearing gradients. Since significant hyperparameter tuning was not needed, around only twenty hyperparameter combinations were evaluated, limited to less than a total of fifty runs. The metrics in Table 6 are based on the evaluation of the model on the validation subsets. For the unseen testing subset, the metrics of the model trained on the joint nuclei/epitheleum/tubule datasets are presented in Table 7.

With regard to image predictions, a few randomly selected images of the test set with the ground-truth segmentation masks (left), juxtaposed with the segmentation masks predicted by the model (right), are presented in Figure 2 below.

### 4.3. Comparison with SOTA

For TUPAC16, compared to the challenge entries presented in [32], the performance of this model is found to be satisfactory, giving it fourth place in the challenge standings. This model’s performance stands out even more considering the technical limitations, the lack of experience and unavailability of domain knowledge. Given that the entries with better F-scores are close to ours, it is possible that an improved process and the improvements outlined below could lead to a higher position in the challenge.

A further fact to be considered is that the other entries were evaluated with an external unpublished testing subset, whereas the presented model was evaluated with a subset of the published dataset, giving a ~20% smaller training subset. Furthermore, some entries used external datasets to further augment their training subset and help with generalization, while this research’s focus was on technical specifications and settings. Lastly, the presented approach used an off-the-shelf, easily set and tuned DL framework instead of custom architectures intertwined with complex non-DL CV procedures.

For JPATHOL, a functioning multi-class instance segmentation model for medical images was prepared that could serve as a baseline for future model development. It showcases how an off-the-shelf commercial product such as Detectron2 can be easily used for niche problems, such as medical imaging, with limited requirements in user expertise or available data. Significant performance gains can be expected by using fully annotated data and exhaustive hyperparameter tuning. Similar problem statements and solutions based on the JPATHOL have not been found in the bibliography.

## 5. Discussion and Conclusions

For TUPAC16, all teams that performed mitosis detection as part of the Tumor Proliferation Assessment Challenge used CNNs [32]. Most teams trained a two-class classification model, with patches centered on a mitotic figure and background patches. On the testing dataset, the model evaluated every pixel location and produced a mitosis probability map that could be further processed to identify mitotic figures and/or produce a mitotic score for a ROI. The neural network architectures applied to this problem vary from relatively shallow with only a few convolutional layers to deep ResNets.

Since mitoses are generally rare events, the mitosis detection problem is very unbalanced. Two main strategies were used to mitigate this, including data augmentation by geometric transformations and hard negative mining. The mitosis detection problem is invariant to rotations, flipping and small translation and scaling, so it can be exploited to create new plausible training samples to enrich the training data. The other strategy was hard negative mining [33], which is a boosting-like technique, where an initial mitosis detection method is trained with random sampling for the background class and then used to detect difficult negative instances that are used to train a second method. In practice, models trained with random sampling for the background class result in many false positives, since all hyperchromatic objects are detected as mitoses. The output of the initial mitosis detection method can be used to sample such difficult background samples and train a second mitosis detection method, which can lead to improvements in mitosis detection accuracy.

The object detection model results can be considered a satisfactory baseline for further fine-tuning and model improvements. The model’s F-score reached fifth place at 0.628, with the best entry at 0.669. Training and inferring with this model are both fast, acceptable models can be obtained within an hour and prediction results are near instantaneous. Average accuracy is also satisfactory with limited data preprocessing and fine-tuning. Even examining the prediction images shows that many false predictions can be interpreted as human-like mistakes, especially acknowledging the fact that part of the ground-truth mitoses may not have been annotated at all, conceivably generating numerous false negatives.

For the three instance segmentation tasks described in JPATHOL, its authors provided some initial processing methodologies to promote further research on their datasets. Further research for the use of similar DL techniques on this dataset could not be found.

For nuclei segmentation, using the procedure outlined and an AlexNet network structure, the authors developed a 5-fold cross-validation set of approximately 100 training and 28 testing images. Qualitatively, the network returns smoother boundaries at 40× magnification rather than at 20×. Quantitatively, the detection rate, i.e., the ability to find nuclei in the image, is very high, with the network identifying 98% of all nuclei at the 40× magnification, and dropout appears to impact the metrics negatively, as presented in Table 8.

For epithelium segmentation, 5 folds of 34 training and 8 test images are preprocessed and passed through the same AlexNet framework as in the previous task. The threshold is used as a hyperparameter for each fold in search of the best possible F-score. Pathologists often treat this task as a higher-level abstraction instead of a pixel level classification, without removing white background pixels. The approach followed by Janowczyk in 2016 [18] can identify smaller regions ignored by pathologists, because they are considered clinically irrelevant. After review of their results by a clinical collaborator, they were found to be suitable for use in conjunction with other classification algorithms, e.g., prognosis prediction. This is one of the first attempts at direct segmentation and quantification of epithelium tissue in breast tissue. The mean F-score for 5-fold cross-validation is 0.84.

For tubule segmentation, each of the 5-fold cross validation sets has about 21 training images and 5 test images. The mean F-score using a threshold of 0.5 was 0.827 ± 0.05. When optimized with a threshold on a per fold basis, this measure rose slightly to 0.836 ± 0.05. Combining all the test sets together, the *p*-value equals 0.33, indicating that there was no significant difference between the expected clinical grade associated with the presented approach and the expert’s ground-truth annotation. Two other state-of-the-art approaches claim 86% accuracy and a 0.845 object-level dice coefficient.

The instance segmentation model can be judged only as a first-stage proof of concept. The data pipeline with preprocessing, training, and inferring is functioning, but the final model performance metrics are mediocre for the three datasets, both jointly and separately. The visual predictions corroborate the numerical performance metrics, demonstrating the model’s difficulty in detecting the epithelium class in particular and any smaller instances in general. Moreover, they present multiple overlapping bounding box predictions from different classes, which should not be a possible outcome with these datasets. A possible explanation for the difference in performance between the object detection and instance segmentation tasks may be the differences in magnification and the texture of the objects, which may affect the robustness of the features that the models are able to extract.

### 5.1. Future Work

A few improvements and suggestions to be considered for further research, concerning both the TUPAC16 and JPATHOL datasets, are outlined below. A significant oversight—due to wrongful assumptions—was the omission of a data augmentation process. As these datasets are invariant to rotation, flipping and scaling, it would be beneficial to incorporate this into the process and enlarge the training subsets. Instead of manually searching in the hyperparameter grid, an automated, random search could help avoid local minima during training, specify a tighter range of hyperparameter values to be tested more thoroughly and possibly achieve better model performance. Furthermore, Detectron2 has many different pretrained CNN models available for testing, so a few could be tested for increased performance or accuracy. Additional exploration could focus on the robustness of the features extracted by the model; data augmentation could assist the model in becoming more generalizable among various similar medical tasks (e.g., mitoses vs. tubule detection) and against variability in the appearance of the tissue to be examined (coloring, brightness, noise, zoom, etc.).

Another problem faced was the variability in tissue appearance and lack of standard staining processes; different staining normalization methods could be tested to diminish these differences, which would help with working on data from different sources and annotators. The annotations of this dataset do not cover all instances of mitosis, which will weaken the model’s training ability and performance. Steps should be taken to extract any other mitosis instances present in the dataset as accurately and thoroughly as possible. This could be undertaken as an unsupervised learning problem with non-DL techniques, e.g., SVM or boosting. A medical expert offering field knowledge would be invaluable in inspecting the dataset and annotations and visually evaluating the annotation extraction process and predictions.

Further suggestions would be to evaluate the training process more closely using libraries such as Tensorboard and adding a feature that rejoins prediction subimages with the rest of the related subimages to form the initial full-resolution image. Explainable AI is also a rather interesting technique, promising to improve DL models by making its internal processes explainable to humans, who can monitor whether the models are trained on specific features correctly.

Concerning the TUPAC16 challenge, the final evaluation can be performed by submitting the highest evaluated model to the organizers who will run it on a separate unseen test set. This is expected to improve the model’s performance, as the current testing subset would be used as additional training data without the risk of overfitting.

In the JPATHOL challenge, we started development directly on all three datasets simultaneously. It is possible that further improvements may be achieved if three separate instance detection models are trained on each of the sub-datasets separately and then joined into one multi-class model. In this manner, the necessary preprocessing for each dataset could be determined more exactly, any deficiencies in the datasets could be rectified earlier and any possible performance loss due to instance class or size could be examined more systematically. Furthermore, for instances such as epithelium and tubule, which, as entities, are much larger compared to individual components, a greater viewing area may be needed to provide sufficient context to make an accurate assessment.

### 5.2. Conclusions

The main objective of this thesis was to gain valuable knowledge and experience in the domain of deep learning, especially where it may be applied to the field of digital pathology. Two separate medical image datasets were chosen for that objective, and two different problems were formulated around them, including one for object detection and one for instance segmentation.

Detectron2, an open source and easily customizable DL framework specialized in computer vision tasks, was selected, which proved to be a solid choice. Its flexibility and tunability allowed for quick training/evaluation cycles, which led to a very competent model for the TUPAC16 challenge. Using that model as a starting point for the JPATHOL instance segmentation task, a functioning multi-class segmentation model was developed. A range of improvements has been suggested that are expected to significantly boost the model’s performance.

In sum, a baseline model for single-class object detection tasks has been successfully developed, as well as a proof-of-concept model for multi-class instance segmentation tasks based on WSI/medical histology slide datasets.

## Figures and Tables

**Figure 1 jpm-12-01444-f001:**
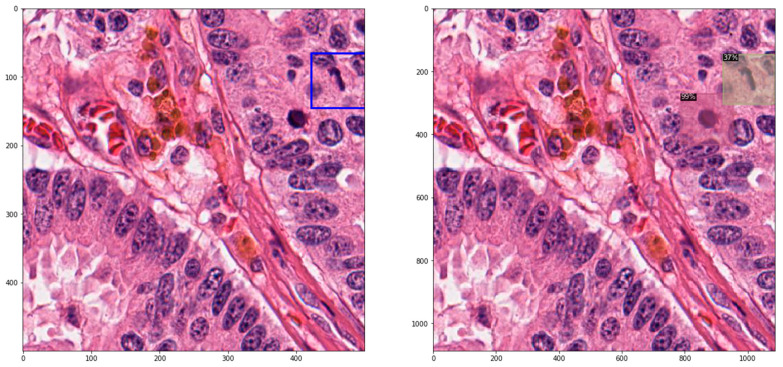
Sample WSIs from the test set with annotated ground truth (**left**) and predictions (**right**).

**Figure 2 jpm-12-01444-f002:**
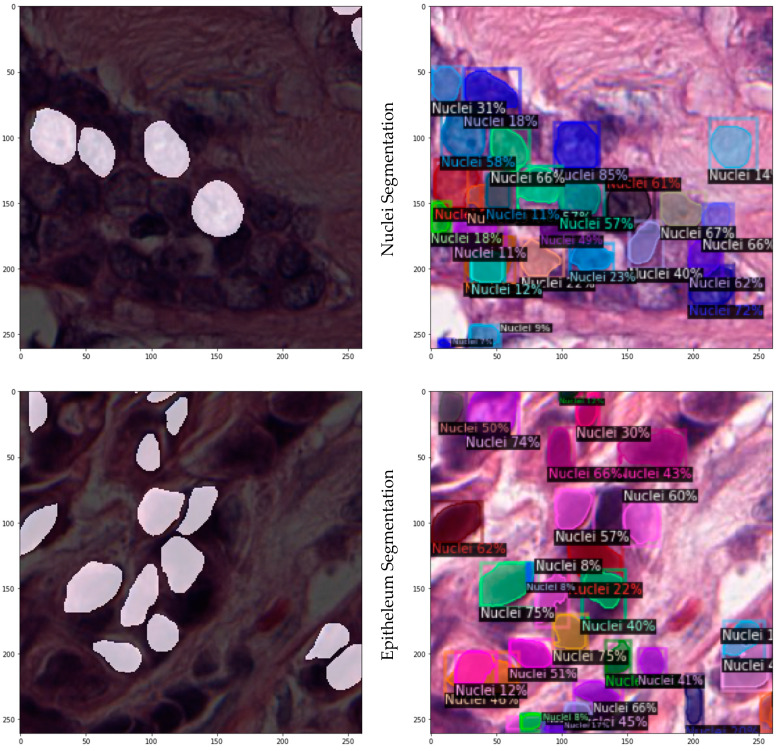
Sample WSIs from the test set with annotated ground truth (**left**) and predictions (**right**).

**Table 1 jpm-12-01444-t001:** Digital pathology task descriptions.

Task	Biological Motivation	Dataset
Nuclei segmentation	Pleomorphism is used in current clinical grading schemes	141 × 2000 × 2000 @40× ROIs of ER+ BCa, containing subset of 12,000 annotated nuclei
Epithelium segmentation	Epithelium regions contribute to identification of tumor infiltrating lymphocytes (TILs)	42 × 1000 × 1000 @20× ROIs from ER+ BCa, containing 1735 regions
Tubule segmentation	Area estimates in high power fields are critical in BCa grading schemes	85 × 775 × 522 @40× ROIs from colorectal cancer, containing 795 delineated tubules

**Table 2 jpm-12-01444-t002:** Alexnet configuration.

Layer	Type	Kernels	Kernel Size	Stride	Activation
0	Input	3	32 × 32		
1	Convolution	32	5 × 5	1	
2	Max pool		3 × 3	2	ReLU
3	Convolution	32	5 × 5	1	ReLU
4	Mean pool		3 × 3	2	
5	Convolution	64	5 × 5	1	ReLU
6	Mean pool		3 × 3	2	
7	Fully connected	64			Dropout + ReLU
8	Fully connected	2			Dropout + ReLU
9	SoftMax				

**Table 3 jpm-12-01444-t003:** AlexNet hyperparameters.

Variable	Setting
Batch size	128
Learning rate	0.001
Learning rate schedule	Adagrad
Rotations	0, 90
Num. iterations	600,000
Weight decay	0.004
Random minor	Enabled
Transformation	Mean-centered

**Table 4 jpm-12-01444-t004:** Detectron2 hyperparameter values.

Hyperparameter	Tuning Range	Starting Value	Optimum Value
Number of workers	(2, 8)	2	8
Images per batch	(2, 8)	2	4
Learning rate	(0.00025, 0.1)	0.00025	0.025
Max iterations	(300, 12,000)	300	9000
Batch size per image	(128, 1024])	128	512
Testing threshold	(0.01, 0.7)	0.7	0.07
Bounding box dimension	(40, 120)	80	80
Subimage split dimension	(224, 1000)	1000	500

**Table 5 jpm-12-01444-t005:** Results for object detection task on TUPAC16.

Images/ Batch	Learning Rate	Max Iterations	RoIHead Batch Size	Split Dimension	Train Time (H:M:S)	AP (%)	STDEV (±%)
4	0.025	600	512	1000 × 1000	0:04:56	43.32	6.58
4	0.025	600	1024	1000 × 1000	0:05:23	40.11	8.65
8	0.025	600	1024	1000 × 1000	0:11:35	43.79	5.69
6	0.010	9000	1024	1000 × 1000	2:02:55	55.74	3.26
4	0.025	9000	512	1000 × 1000	1:13:18	58.11	2.96
4	0.025	9000	512	500 × 500	1:11:49	65.02	4.05

**Table 6 jpm-12-01444-t006:** Results for instance segmentation task on JPATHOL on validation set.

Dataset	AP (%)	STDEV (±%)
Nuclei	19.53	1.68
Epithelium	5.15	3.21
Tubule	35.22	5.16
Nuc/Epi/Tub	8.03	2.09

**Table 7 jpm-12-01444-t007:** Segmentation results on test set.

AP	AP50	AP75	Aps	Apm	Apl	AP-Nuc	AP-Epi	AP-Tub
14.41	23.72	15.31	5.16	22.42	17.67	15.26	2.06	25.91

**Table 8 jpm-12-01444-t008:** Nuclei segmentation results.

Method	Detection	F-Score
20×	0.95	0.80
20× + dropout	0.90	0.79
40×	0.98	0.83
Baseline model—40×	0.14	0.220

## Data Availability

Not applicable.

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
