# Peer review of "Image Analysis in Digital Pathology Utilizing Machine Learning and Deep Neural Networks [Author-notes fn1-jpm-12-01444]"

_jpm, 2022, doi:10.3390/jpm12091444_

Round 1
Reviewer 1 Report
The article is very poorly written. The readability is difficult. The proposed research contributions are not convincing. Needs greater improvement in all sections, especially the experimental results section and contributions.
Author Response
To the editor and reviewers of the Journal of Personalized Medicine
25 August 2022
Subject: Revised submission of Manuscript JPM-1789553
We submit the revised manuscript, entitled:
“Image Analysis in Digital Pathology utilizing Machine Learning & Deep Neural Networks”.
We would like to thank you for your very constructive comments on the initial submission of our manuscript. We understand that our submission had some issues and we would like to thank you for pointing them out.
We hope that the revised version of our manuscript addresses all your comments and answers your questions. Please find below the list of your comments and our responses. Kindly note that our corrections and additions appear in the revised manuscript in red color.
Sincerely yours,
Paris Amerikanos
(On behalf of the authors)
Responses
Reviewer #1:
The article is very poorly written. The readability is difficult. The proposed research contributions are not convincing. Needs greater improvement in all sections, especially the experimental results section and contributions. .
Answer:
We would like to thank you for reviewing our report. In defense of our work, a native English speaker with a degree in linguistics has reviewed the readability of our report before submission. Of course, we realize there may still be areas of more technical nature that may require further improvements; if you would be so kind to point out some specific parts that you believe are not up to par, that would benefit us tremendously? Furthermore, would it be possible to request in what ways our contributions are not considered convincing? The area of Medical AI is rapidly growing, and there has been extensive work in the CV/ML field with applications in all sorts of pathology tasks, as presented in the Related Work section and our references. We would respectfully like to request further elaboration on the request for improvement in all sections.
Reviewer 2 Report
Dear authors, thank you very much for the opportunity to review this manuscript.
To me, there are few issues to be addressed to improve the quality of the publication:
- I disagree with the interpretation of the term "theragnosis", and seems a bit forced in the context. please discuss this.
- on page 19 it appears 7Error! Reference source not found. Please thoughly proofread the manuscript before resubmitting.
-on page 3 you state the CAD software did not diminish the role of the radiologist. It is a bold statement, and needs to be supported by references, as there is a wide debate over this topic.
in general, the manuscript lacks proper references over key topics such as demographics of BC.
Author Response
To the editor and reviewers of the Journal of Personalized Medicine
25 August 2022
Subject: Revised submission of Manuscript JPM-1789553
We submit the revised manuscript, entitled:
“Image Analysis in Digital Pathology utilizing Machine Learning & Deep Neural Networks”.
We would like to thank you for your very constructive comments on the initial submission of our manuscript. We understand that our submission had some issues and we would like to thank you for pointing them out.
We hope that the revised version of our manuscript addresses all your comments and answers your questions. Please find below the list of your comments and our responses. Kindly note that our corrections and additions appear in the revised manuscript in red color.
Sincerely yours,
Paris Amerikanos
(On behalf of the authors)
Responses
Reviewer #2
Dear authors, thank you very much for the opportunity to review this manuscript.
To me, there are few issues to be addressed to improve the quality of the publication:
- I disagree with the interpretation of the term "theragnosis", and seems a bit forced in the context. please discuss this.
- on page 19 it appears 7Error! Reference source not found. Please thoroughly proofread the manuscript before resubmitting.
- on page 3 you state the CAD software did not diminish the role of the radiologist. It is a bold statement, and needs to be supported by references, as there is a wide debate over this topic.
- in general, the manuscript lacks proper references over key topics such as demographics of BC.
Answer:
Thank you for your comments – we have attempted to address them as requested:
- Theragnosis may indeed not be a fitting term, so it was changed to “Personalized therapy” which we believe may better fit this section on p.3 under “The Concept of Digital Pathology”.
- Missing references to two tables on p.19 under “JPATHOL”, and various other typos have been fixed.
- The statement on p.3 about the “Role of the radiologist” has been removed as it is, indeed, debatable, but not the central point of this study.
- Particular care had been taken to refer as many of the sources used as possible. A missing reference about the BC demos was added (Shah, 2020) on p.4 under “Role of the pathologist”.
Reviewer 3 Report
This paper presented a study investigating the potential of deep learning-based methods for digital pathology (DP). Specifically, the author explored using a state-of-the-art deep learning framework (Detectron2) to perform object (mitoses) detection on the TUPAC16 dataset and instance (nuclei, epithelium, and tubule) segmentation on the JPATHOL dataset. The experimental results showed that, for the object detection task, the deep learning model can effectively achieve an acceptable accuracy with high efficiency. However, for the instance segmentation task, the performance of the deep learning model is still at a proof-of-concept level.
This paper generally is well-written, and the study revealed the status of current deep learning techniques in the DP applications. Overall, I have a positive opinion of this paper but still have some concerns, which should be carefully addressed before I can make the final recommendation.
Major comments:
1) Section 2 “Related Work and Background Information” -> subsection “State-of-The-Art Computer Based Methodologies and Tools for Digital Pathology”: The author categorized “computer vision” as a kind of methodology. However, I think computer vision should be more like a subject or a research area (just like the subject of natural language processing (NLP)) rather than a methodology. Machine learning is the mainstream methodology we often used for solving computer vision problems.
2) Section 2 “Related Work and Background Information” -> subsection “Deep Learning in Computer Vision”: The author categorized deep learning as a branch of computer vision, which could be improper. It could be better to say that deep learning is a branch of machine learning. Machine learning can be divided into two categories of methods. One type is the conventional machine learning methods, which rely on handcraft features. The other type is deep learning, which utilizes the self-learned hierarchical features.
3) Section 5 “Discussion and Conclusion”: The author is suggested to explain or analyze why the deep learning framework can achieve a relatively acceptable accuracy in the object (mitoses) detection task but just a proof-of-concept performance in the instance (nuclei, epithelium, and tubule) segmentation task? What factors could be the cause of such a difference?
Minor comments:
1) Section 1, second paragraph: There are two dots after the last sentence in this paragraph.
2) Section 2, subsection of “Role of the pathologist”, “Breast cancer accounts for 25% of all cancer cases in women worldwide. In 2018 it resulted in two million new cases and 627,000 deaths.”: Is there any reference for this statement?
3) Section 2, subsection of “Machine Learning”, “Supervised learning relates inter alia to: OCR, speech recognition, …”: Please specify the full name of “OCR”. Is it optical character recognition?
4) Section 2, subsection of “Machine Learning”, a single line of “Feature Extraction and Selection”: Is this single line a title of a subsection just like the line of “Machine Learning” before? If so, please fix the format.
5) Section 3.1, a single line of “Deep Learning Framework Specifications”: The same format issue as my minor comment #4.
6) Section 3.2.2 “Instance Segmentation on JPATHOL”, “The ground truth annotations are usually performed by an expert …”: The “ground truth” (noun) should be “ground-truth” (adjective) here. Same issue for the following contents.
7) Section 3.2.2 “Instance Segmentation on JPATHOL”, paragraph of “NUCLEI SEGMENTATION”, “… the random sampling from the complement image may or may return unmarked nuclei belonging to …”: “may or may” -> “may or may not”
8) Section 3.2.4 “Parameterization & Training”: The author introduced that “Finding the optimum values is a process that can be performed either manually or automatically.” However, we still don’t know in which way this work determined the hyper-parameters?
9) Page 14: The command lines printed on this page (“Average Precision (AP) IoU=0.50:0.95 | area= all | AP …”) should be re-organized in a table.
10) Table 5: I am confused about the two columns named “Images / batch” and “Batch / image”. What’s the meaning of them?
11) Page 19: There is an “Error! Reference source not found” on this page. Please fix it. Same issue for the following contents.
12) Table 7: It is unclear which model from Table 6 is tested to get the results reported in Table 7. And also, the table caption “NUCLEI SEGMENTATION RESULTS” may be improper since the task does not only include the nuclei segmentation but also the epithelium&tubule segmentation.
13) Figure 4: The author is suggested to show an example case of tubule segmentation. Currently, we can only see examples of nuclei segmentation and epithelium segmentation.
14) Section 5, “Discussion and Conclusion”, “Qualitatively, the network returns crisper results at 40× magnification …”: The author used the term "crisp" to describe the segmentation results. I am confused about the meaning of this. Does that mean the segmentation results have sharp boundaries with high contrast against the background? Is it a professional term often used in DP? If not, I suggest the author replace "crisp results" with "sharp boundaries" since the latter is easier for understanding.
15) Table 8: It is unclear about the meaning of the last row of Table 8. What’s the model of “Present model - 40x”?
Author Response
To the editor and reviewers of the Journal of Personalized Medicine
25 August 2022
Subject: Revised submission of Manuscript JPM-1789553
We submit the revised manuscript, entitled:
“Image Analysis in Digital Pathology utilizing Machine Learning & Deep Neural Networks”.
We would like to thank you for your very constructive comments on the initial submission of our manuscript. We understand that our submission had some issues and we would like to thank you for pointing them out.
We hope that the revised version of our manuscript addresses all your comments and answers your questions. Please find below the list of your comments and our responses. Kindly note that our corrections and additions appear in the revised manuscript in red color.
Sincerely yours,
Paris Amerikanos
(On behalf of the authors)
Responses
Reviewer #3:
This paper presented a study investigating the potential of deep learning-based methods for digital pathology (DP). Specifically, the author explored using a state-of-the-art deep learning framework (Detectron2) to perform object (mitoses) detection on the TUPAC16 dataset and instance (nuclei, epithelium, and tubule) segmentation on the JPATHOL dataset. The experimental results showed that, for the object detection task, the deep learning model can effectively achieve an acceptable accuracy with high efficiency. However, for the instance segmentation task, the performance of the deep learning model is still at a proof-of-concept level.
This paper generally is well-written, and the study revealed the status of current deep learning techniques in the DP applications. Overall, I have a positive opinion of this paper but still have some concerns, which should be carefully addressed before I can make the final recommendation.
Major comments:
1) Section 2 “Related Work and Background Information” -> subsection “State-of-The-Art Computer Based Methodologies and Tools for Digital Pathology”: The author categorized “computer vision” as a kind of methodology. However, I think computer vision should be more like a subject or a research area (just like the subject of natural language processing (NLP)) rather than a methodology. Machine learning is the mainstream methodology we often used for solving computer vision problems
2) Section 2 “Related Work and Background Information” -> subsection “Deep Learning in Computer Vision”: The author categorized deep learning as a branch of computer vision, which could be improper. It could be better to say that deep learning is a branch of machine learning. Machine learning can be divided into two categories of methods. One type is the conventional machine learning methods, which rely on handcraft features. The other type is deep learning, which utilizes the self-learned hierarchical features.
3) Section 5 “Discussion and Conclusion”: The author is suggested to explain or analyze why the deep learning framework can achieve a relatively acceptable accuracy in the object (mitoses) detection task but just a proof-of-concept performance in the instance (nuclei, epithelium, and tubule) segmentation task? What factors could be the cause of such a difference?
Minor comments:
1) Section 1, second paragraph: There are two dots after the last sentence in this paragraph.
2) Section 2, subsection of “Role of the pathologist”, “Breast cancer accounts for 25% of all cancer cases in women worldwide. In 2018 it resulted in two million new cases and 627,000 deaths.”: Is there any reference for this statement?
3) Section 2, subsection of “Machine Learning”, “Supervised learning relates inter alia to: OCR, speech recognition, …”: Please specify the full name of “OCR”. Is it optical character recognition?
4) Section 2, subsection of “Machine Learning”, a single line of “Feature Extraction and Selection”: Is this single line a title of a subsection just like the line of “Machine Learning” before? If so, please fix the format.
5) Section 3.1, a single line of “Deep Learning Framework Specifications”: The same format issue as my minor comment #4.
6) Section 3.2.2 “Instance Segmentation on JPATHOL”, “The ground truth annotations are usually performed by an expert …”: The “ground truth” (noun) should be “ground-truth” (adjective) here. Same issue for the following contents.
7) Section 3.2.2 “Instance Segmentation on JPATHOL”, paragraph of “NUCLEI SEGMENTATION”, “… the random sampling from the complement image may or may return unmarked nuclei belonging to …”: “may or may” -> “may or may not”
8) Section 3.2.4 “Parameterization & Training”: The author introduced that “Finding the optimum values is a process that can be performed either manually or automatically.” However, we still don’t know in which way this work determined the hyper-parameters?
9) Page 14: The command lines printed on this page (“Average Precision (AP) IoU=0.50:0.95 | area= all | AP …”) should be re-organized in a table.
10) Table 5: I am confused about the two columns named “Images / batch” and “Batch / image”. What’s the meaning of them?
11) Page 19: There is an “Error! Reference source not found” on this page. Please fix it. Same issue for the following contents.
12) Table 7: It is unclear which model from Table 6 is tested to get the results reported in Table 7. And also, the table caption “NUCLEI SEGMENTATION RESULTS” may be improper since the task does not only include the nuclei segmentation but also the epithelium&tubule segmentation.
13) Figure 4: The author is suggested to show an example case of tubule segmentation. Currently, we can only see examples of nuclei segmentation and epithelium segmentation.
14) Section 5, “Discussion and Conclusion”, “Qualitatively, the network returns crisper results at 40× magnification …”: The author used the term "crisp" to describe the segmentation results. I am confused about the meaning of this. Does that mean the segmentation results have sharp boundaries with high contrast against the background? Is it a professional term often used in DP? If not, I suggest the author replace "crisp results" with "sharp boundaries" since the latter is easier for understanding.
15) Table 8: It is unclear about the meaning of the last row of Table 8. What’s the model of “Present model - 40x”?
Answer:
Thank you for the time you took to examine our article so closely. Your input was most valuable to improve as required, and we attempted to address every issue raised.
Major comments:
- While we completely agree with your statement, we believe we have made no statement or suggestion that CV is a methodology, except, perhaps, the phrase where we mention CV feature-based methods, but that may have been misconstrued. For clarity, we have added a mention on p.4 under “Computer Vision” that CV is a research field, as proposed.
- As with comment #1, there is full agreement with your suggestion, but we believe we have made no mention that DL is a CV branch. For clarity, we have added “DL as a branch of ML” on p.6 under “Feature Extraction and Selection”, as proposed.
- That was an interesting question raised, and we have attempted to summarily (but not-exhaustively) address it on p.23 under the “Discussion and Conclusion” section. Our experiment was not set up to give a clear answer to this question, but this was added on p.24 under “Future Work” as an intriguing exploratory task to be examined.
Minor comments:
- Removed the second dot in Section 1, second paragraph.
- A missing reference about the BC demos was added (Shah, 2020) on p.4 under “Role of the pathologist”.
- In Section 2 under “Machine Learning”, OCR was expanded to Optical Character Recognition.
- In Section 2 under “Machine Learning”, fixed the format of the subsection title.
- In Section 3.1 under “Deep Learning Framework Specifications”, fixed the format of the subsection title.
- In Section 3.2.2 under “Instance Segmentation on JPATHOL”, changed an instance of “ground truth” (noun) to “ground-truth” (adjective) here. No other instances found.
- In Section 3.2.2 under “Instance Segmentation on JPATHOL”, fixed typo “may or may” to “may or may not”.
- In Section 3.2.4 under “Parameterization & Training”, we added a clarification that our hyperparameter tuning was done in a manual fashion. The specific implementation of Detectron2, due to its lengthy training sessions and the way the framework consumed but did not release the GPU’s memory after each session, did not allow for automatic search through the hyperparameter grid, which would have been preferrable for performance, speed and ease.
- On Page 14 the text box that outlined the various object detection metrics (“Average Precision (AP) IoU=0.50:0.95 | area= all | AP …”) was deemed obsolete as it offered no clear insights to the rest of the experiment metrics.
- In Table 5 “Images / batch” is the usual batch-size term used for loading data into DL models during training, whereas “Batch / image” is a term that refers to the ROIhead batch size , which is specific to the object detector architecture of Detectron2/MaskRCNN. We changed the column header for clarity.
- Missing references to two tables on p.19 under “JPATHOL”, and various other typos have been fixed.
- On Page 19 under “JPATHOL” an addition was made to clarify that the metrics from Table 7 belong to the 4th model in Table 6, the model trained on the joint Epi/Nuc/Tub datasets. We also removed the “Nuclei” tag from the title of Table 7 as, indeed, it includes results for all three classes.
- In Figure 4 we present examples of all three classes (Epi/Nuc/Tub). We fixed a typo where we mislabeled the example case of tubule segmentation.
- In Section 5 under “Discussion and Conclusion”, the term "crisp" was changed to smooth. Indeed, a mask cannot be crisp, and it was a counterintuitive description. Instead, we chose “smooth” to describe the border of the mask, as it showed less noise and “waviness” along the length of its boundary.
- In Table 8 we changed the name of Present model to Baseline, where Baseline is the model that was trained on the dataset using the hyperparameters from the object detection/TUPAC task trained in the first part.